# Concept for Markerless 6D Tracking Employing Volumetric Optical Coherence Tomography

**DOI:** 10.3390/s20092678

**Published:** 2020-05-08

**Authors:** Matthias Schlüter, Lukas Glandorf, Martin Gromniak, Thore Saathoff, Alexander Schlaefer

**Affiliations:** Institute of Medical Technology, Hamburg University of Technology, 21073 Hamburg, Germany; matthias.schlueter@tuhh.de (M.S.); lukas.glandorf@tuhh.de (L.G.); martin.gromniak@tuhh.de (M.G.); thore.saathoff@tuhh.de (T.S.)

**Keywords:** markerless tracking, tracking systems, optical coherence tomography, motion estimation

## Abstract

Optical tracking systems are widely used, for example, to navigate medical interventions. Typically, they require the presence of known geometrical structures, the placement of artificial markers, or a prominent texture on the target’s surface. In this work, we propose a 6D tracking approach employing volumetric optical coherence tomography (OCT) images. OCT has a micrometer-scale resolution and employs near-infrared light to penetrate few millimeters into, for example, tissue. Thereby, it provides sub-surface information which we use to track arbitrary targets, even with poorly structured surfaces, without requiring markers. Our proposed system can shift the OCT’s field-of-view in space and uses an adaptive correlation filter to estimate the motion at multiple locations on the target. This allows one to estimate the target’s position and orientation. We show that our approach is able to track translational motion with root-mean-squared errors below 0.25 mm and in-plane rotations with errors below 0.3°. For out-of-plane rotations, our prototypical system can achieve errors around 0.6°.

## 1. Introduction

Optical tracking is an established approach in a wide range of applications to estimate or compensate the motion of a target. One particularly important field is the navigation of medical procedures, for example, in radiation therapy, neurosurgery, laparoscopy, or orthopedic surgery. In some fields, we mainly consider the tracking of rigid structures, which can be described by their pose; i.e., position and orientation in space. A classic example is the human skull during surgeries [1]. Optical markers can be used to determine the pose with respect to some reference frame. However, marker placement is critical for the accuracy of the approach [2]. Accurate head tracking is also fundamental in cranial radiosurgery. If the patient moves his head during the treatment, we irradiate healthy areas instead of the tumor. Therefore, we have to detect any larger motion to stop treatment and avoid damage. To reduce the likelihood of such situations, different approaches to immobilize the head can be employed; for example, stereotactic frames or thermoplastic masks. These approaches, however, reduce patient comfort and can make patient positioning more difficult. Additionally, there is always some residual motion in the case of masks. For these reasons, head tracking is an important approach, especially because systems like the CyberKnife (Accuray) allow one to dynamically adjust the direction of the treatment beams. However, this requires continuous tracking of the head’s pose with high accuracy and low latency. Markerless approaches to this task employ, for example, structured light [3] or time-of-flight cameras [4]. Extending point cloud models with additional information, such as tissue thickness estimation, has been proposed [5]. Another application for head tracking in medicine is robotic transcranial magnetic stimulation (rTMS). In rTMS, we use a robot to place a coil closely to the patient’s head. This coil non-invasively stimulates a brain region with magnetic pulses. Tracking the head’s pose is fundamental to continuously adapting the coil pose and ensuring steady stimulation of the correct brain region.

Another class of targets constitutes moving and deforming structures within the patient. In abdominal radiosurgery, liver or prostate motion has to be tracked for correct irradiation. If respiration is the cause of the motion, external optical markers can serve as a surrogate to estimate the internal motion [6]. Direct optical tracking of organs frequently occurs in minimally invasive surgeries, such as laparoscopy. While markerless instrument tracking [7] can typically exploit clear and known geometries, this is not the case for organs. Their borders might not be within the imaging range, their shapes and appearances can substantially vary between patients, and textural features on their surfaces can be rare and sensitive to moisture, illumination, and other external conditions. Recently, there has been an increasing interest in integrating augmented reality (AR) and mixed reality (MR) in the operating rooms, especially due to the release of devices such as HoloLens (Microsoft) [8]. While virtual reality (VR) is mainly increasing the efficiency of training or planning applications, AR and MR promise to provide information more concise and intuitive to the surgeon during actual interventions [9,10,11]. Instead of observing visualizations and data on several monitors, information is directly fused and visualized by superimposing it to the surgeon’s current view onto the patient. Therefore, they require one to solve several processing tasks, such as object detection, segmentation, 3D scene reconstructions, registration with previous data, localization, and pose estimation. Furthermore, all these tasks have to be solved online with small delays in order to provide an acceptable perception. This emphasizes that efficient solutions for high-accuracy tracking in difficult environments are still an important topic today.

In summary, highly accurate tracking is an important issue for various applications involving navigation, robotics, and VR/AR. If we want to overcome the limitations of markers, most approaches are limited to targets with clear or known textures or borders. This is problematic in cases of unknown smooth surfaces and under difficult conditions, like limited space. For these reasons, we propose to consider the imaging modality optical coherence tomography (OCT) for markerless tracking. The basis of OCT are one-dimensional interferometric measurements employing near-infrared light, typically at a center wavelength which experiences rather little absorption in water. Thereby, it is possible to acquire depth profiles (A-scans) containing information about reflecting structures below the surface, which are not accessible with superficially scanning modalities. In general, the imaging range of an A-scan covers only few millimeters, but it provides micrometer-scale resolution and acquisition rates above 1 MHz [12]. Scan heads, typically utilizing galvanometer mirrors (galvos), allow one to laterally translate the OCT beam to obtain sequences of A-scans representing cross-sectional (B-scans) or volumetric images (C-scans). With the latest systems, even high-resolution C-scans can be acquired at video rate [13] and with low motion artifacts [14].

OCT imaging is an established technique in ophtalmology, but it is also increasingly being used in other fields, for instance, in intravascular diagnosis in cardiology [15] or dermatology [16]. OCT has also been studied to guide interventions like cochleostomy [17] or to monitor medical lasers [18,19]. For intra-operative applications, integration of OCT scanning into surgical tools has been proposed [20,21]. OCT has been studied for image registration in the contexts of tracking of artificial markers [22] and video stabilization [23]. Additionally, different deep-learning approaches have been evaluated recently for learning known marker geometries [24] or evaluating optical flow [25]. However, the constantly limiting aspect when using OCT imaging is the very small field-of-view (FOV). Although experimental systems allowing for larger-scale OCT imaging by different techniques have been described [26,27,28], the FOV of commercial systems is still limited to some millimeters in every direction. In clinical applications, this often leads to using a second modality to guide the actual OCT imaging; for example, angiography in cardiology for catheter placement. To overcome the limited size of the FOV, different approaches have been studied. They usually involve moving the OCT scan head—for example, by integration into a motorized microscope [29] or by mounting it to a robot [30], and some kind of volume stitching [31]. As an alternative which does not require moving the scanning system, we recently proposed a setup which uses a second pair of galvos to laterally move the FOV to different locations [32]. By additionally using a motorized reference arm, the FOV can also be shifted axially. Besides volume stitching, this setup also allows one to compensate for translational motion of a target by dynamically adjusting the FOV position and to recover its trajectory [33]. However, this approach is so far limited to 3D tracking.

In this work, we describe a concept for employing OCT to setup a generic tracking system considering the following key specifications. First, the system should be able to estimate trajectories of 6D motion, i.e., translations and rotations of a rigid target, in a range that substantially exceeds the very limited FOV size. Second, it realizes fully markerless tracking; i.e., no target-specific information is required. Third, targets can be tracked without having their borders in the FOV or even without having a clearly textured surface at all by exploiting OCT’s sub-surface information.

## 2. Material and Methods

In this section, we first describe the hardware setup of our system, which consists of an OCT device and a setup to dynamically reposition the OCT’s FOV. Repositioning the FOV allows one to follow a moving target. We first describe a single-template concept and then propose a multi-template concept which allows one to estimate 6D motion by tracking multiple points on a target. For tracking of the different templates, we employ a MOSSE filter. Finally, we describe in this section a calibration procedure to obtain a Cartesian coordinate frame and the experimental setup for our experiments and evaluations. The overall system is sketched in Figure 1.

### 2.1. Hardware Setup

For imaging, we employ a commercially available swept-source OCT device (OMES, Optores), which consists of an FDML laser, an imaging module, and a scan head. The device has a center wavelength of 1315 nm and provides an A-scan rate of 1.59 MHz, where each A-scan is reconstructed from 2432 raw data samples. By closing a shutter in the sample arm, we estimate the background signal and subtract it from the raw data. The non-linearity of the laser’s sweep is estimated by branching off a recalibration arm from the reference arm and processing the recalibration signal with a correction algorithm provided by the manufacturer’s SDK. Image reconstruction is fully implemented in CUDA to allow for efficient GPU processing. For acquiring C-scans consisting of 32 × 32 axial A-scans, we achieve a C-scan rate of 831 Hz by using a scan head with one resonant galvanometer mirror. We use a lateral scan area of approximately 2.5 mm × 2.5 mm and crop the A-scans to 480 pixels which cover approximately 3.5 mm axially.

We placed a pair of galvos in front of the scan head, which allows one to shift the whole FOV laterally (Figure 2a). In front of these galvos, we have a lens with a focal distance of 30 mm defining the working distance between our system and the target to be tracked. Further, its diameter of 50.8 mm limits the lateral positioning range to a disk. To shift the OCT’s FOV along its axial direction, we employ the motorized reference arm shown in Figure 2b. It consists of a fixed mirror and a retroreflector on a linear guide, which is driven by a stepper motor via a belt. When we move the retroreflector, the FOV is shifted by twice the distance. While this reference arm could be used to realize rather large axial shifts, we only consider shifts in this work with similar lateral and axial ranges.

In summary, our setup allows one to shift the FOV in 3D space by selecting a three-dimensional vector $m={\left(m_{x},m_{y},m_{z}\right)}^{⊺}$ of motor steps, where $m_{x}$ and $m_{y}$ refer to lateral shifting by the two galvos and $m_{z}$ to axial shifting by the motorized reference arm. The motor and galvos are controlled by a microcontroller which communicates with our measurement PC. The PC has a digitizer board (ATS9373, Alazartech) for OCT data acquisition and two GPUs (GTX Titan X and GTX 980 Ti, Nvidia). We use one of the GPUs for image reconstruction, and the second one is available for further data processing; i.e., implementation of a tracking algorithm.

### 2.2. Single-Template Tracking

In order to track a moving target and to adjust the FOV position accordingly, we generally employ a template-based approach. Given a template C-scan $V^{0}\left(x,y,z\right)$ and a later C-scan $V^{i}\left(x,y,z\right)$, an image processing algorithm determines the translation $t\left(V^{0},V^{i}\right)={\left(\Delta x,\Delta y,\Delta z\right)}^{⊺}$ measured in voxels. This translation is converted into a new vector of motor steps $m^{i}$ as
(1)$$\begin{array}{cc} m^{i}= & m^{i-1}+\Delta m^{i}\\ = & m^{i-1}+\mathrm{diag}\left(p_{xy},p_{xy},p_{z}\right)\;\cdot \;C\;\cdot \;t\left(V^{0},V^{i}\right) \end{array}$$
where *C* is a matrix converting voxels into motor steps and $p_{xy}$ and $p_{z}$ are proportional gains for the movement of the galvos and the stepper motor, respectively. We assumed that *C* is a diagonal matrix—i.e., each motor axis is exactly identified with one OCT axis—which turned out to be sufficient for the feedback loop in our setup. We estimated the lateral conversion factors to be 2.5 steps/voxel and the axial conversion to be 0.7 steps/voxel. Note that the axial voxel spacing is substantially smaller than the lateral.

### 2.3. Multi-Template Tracking

The pose of a rigid body can be determined from the locations of three non-collinear points. Therefore, to estimate the orientation in addition to the positions of our moving targets, we simultaneously track multiple templates which are spatially separated. We consider *N* templates and their corresponding initial FOV positions $m_{n}^{0}$, n=1,…N, and arrange them on a circle with radius *r* around a suitable center point $c={\left(x^{0},y^{0},z^{0}\right)}^{⊺}$, as shown in Figure 3. Therefore, we have
(2)$$m_{n}^{0}=c+\left(\begin{array}{c} r\;\cdot \;\cos\left(2\pi \frac{n-1}{N}\right)\\ r\;\cdot \;\sin\left(2\pi \frac{n-1}{N}\right)\\ 0 \end{array}\right)$$
as the initial motor positions. For targets which are not sufficiently flat within radius *r*, we could also define the templates at different depths. However, we assume for now that this is not necessary.

We track these *N* templates sequentially. In iteration *i* for template *n*, we acquire a C-scan $V_{n}^{i}$ at position $m_{n}^{i-1}$; compute its translation with respect to $V_{n}^{0}$ and convert it to motor steps $\Delta m_{n}^{i}$; and determine the next position $m_{n}^{i}$ according to Equation (1). Afterwards, we do the same steps for the next template. However, we can pipeline these operations as sketched in Figure 4 to increase the tracking rate. Note that the most time-consuming part in our setup is moving the FOV axially by changing the length of the reference arm, especially if the employed stepper motor has to change its direction of travel.

We do not consider $m_{n}^{i}$ for tracking directly, but only use it to adjust the FOV position. Considering Equation (Equation 1), we see that the appropriate FOV position for $V^{i}$ is rather
(3)$${\tilde{m}}_{n}^{i}=m_{n}^{i-1}+C\;\cdot \;t\left(V_{n}^{0},V_{n}^{i}\right);$$
i.e., without the proportional gains. As a timestamp for this position, we use the acquisition timestamp of $V^{i}$.

### 2.4. Tracking Algorithm

In previous work [33], we showed that the phase correlation method is a suitable and efficient tracking algorithm for the single-template setup with purely translational motion. It employs cross-correlation to estimate the translation between a template C-scan and a new template C-scan. By using a Fourier-domain approach, phase correlation has very low computational cost.

However, if we also observe substantial rotations, which we cannot compensate by shifting the FOV in our system, then we cannot expect this simple template-based approach to succeed. The rotation has a severe impact on the appearance of the target in the C-scan, and matching with the template C-scan will quickly fail due to the decaying similarity. More recently developed adaptive correlation filters extend the Fourier domain approach of the phase correlation method to gradually alter the filter’s target representation and allow target tracking through appearance changes [34]. We chose to use the MOSSE (minimum output sum of squared error) filter [35] that offers high computational speed, online filter adaptation, and a straightforward extension from two-dimensional to volumetric image data.

Following Bolme et al. [35], we summarize the key concepts of the MOSSE filter for three-dimensional images. It typically does not employ a single template image f(x), $x\in R^{3}$, but constructs an optimal filter H(k) in the Fourier domain from *T* training images $f_{j}\left(x\right)$. This reduces the noise and makes the filter more robust and stable. Let $F_{j}\left(k\right)$ be the three-dimensional Fourier transform of $f_{j}\left(x\right)$. The complex conjugate of the filter H(k) is then constructed from the training images as
(4)$$H^{*}\left(k\right)=\frac{\sum_{j=1}^{T}G_{j}\left(k\right)\;\cdot \;F_{j}^{*}\left(k\right)}{\sum_{j=1}^{T}F_{j}\left(k\right)\;\cdot \;F_{j}^{*}\left(k\right)+\lambda },$$
where λ is a small regularization constant. $G_{j}\left(k\right)$ is the Fourier transform of a Gaussian function
(5)$$g_{j}\left(x\right)=\frac{1}{{\left(\sqrt{2\pi }\sigma \right)}^{3}}\exp\left(-\frac{{\left(s_{j}-x\right)}^{⊺}\left(s_{j}-x\right)}{2\sigma^{2}}\right)$$
which is shifted by $s_{j}$ and represents a smoothed version of the desired filter output. The vector $s_{j}$ represents the translation which the filter should determine for input $f_{j}\left(x\right)$; i.e., the training images do not have to be centered in this algorithm. During tracking, we then obtain the translation t of a new image by transforming the image to the Fourier domain, multiplying it with $H^{*}\left(k\right)$, and applying the inverse Fourier transform to the product. The location of the highest intensity in this result corresponds to the translation of the new image with respect to the filter. The advantage of allowing for arbitrarily shifted images for training becomes even more apparent when considering updating the filter during tracking to adapt to appearance changes. The filter can be updated after each evaluation of a new image. When we evaluate the *i*-th image $f^{i}\left(x\right)$ during tracking, we obtain its translation $t^{i}$ from the current filter $H^{i-1}\left(k\right)$ as described above. Note that we use superscript indices for the image sequence during actual tracking and subscript indices for the training images. The determined translation $t^{i}$ defines the target output $g^{i}\left(x\right)$ of the *i*-th image, and the updated filter is recursively updated as
(6)$$H^{i*}\left(k\right)=\frac{A^{i}\left(k\right)}{B^{i}\left(k\right)}$$
with nominator
(7)$$A^{i}\left(k\right)=\eta \;\cdot \;\left(G^{i}\left(k\right)\;\cdot \;F^{i*}\left(k\right)\right)+\left(1-\eta \right)\;\cdot \;A^{i-1}\left(k\right),$$
denominator
(8)$$B^{i}\left(k\right)=\eta \;\cdot \;\left(F^{i}\left(k\right)\;\cdot \;F^{i*}\left(k\right)+\lambda \right)+\left(1-\eta \right)\;\cdot \;B^{i-1}\left(k\right),$$
and learning rate η∈[0,1]. The higher η, the more influence the new image has on the filter. Updating templates always leads to a critical trade-off [36]. On the one hand, it allows for compensating stronger changes in appearance, but on the other hand, it also increases the likelihood of a drift; i.e., the filter no longer tracks the original target after some time but a structure at another location.

In our implementation, we use ten training images initially acquired at the same position. We use σ=2 for the Gaussians and further set λ=0.001. We update the filter each time we evaluate a new image and compare different learning rates η. The algorithm is implemented in CUDA for processing on a GPU.

### 2.5. Cartesian Calibration

Neither the C-scans nor the motor positions form a Cartesian coordinate frame with physical units. Therefore, we need a calibration to obtain proper coordinates, especially to calculate meaningful orientations from the motor positions. For this purpose, we can use a 3D motion stage and single-template tracking with the phase correlation algorithm to acquire calibration data [33]. We move a marker slowly to different positions within a volume-of-interest with active tracking. Thereby, we obtain *M* pairs of positions $p^{i}$ of the motion stage and motor positions $m^{i}$ of the tracking system. We can firstly determine an affine transformation matrix $A\in R^{3\times 4}$ by solving
(9)$$\min_{A}\sum_{i=0}^{M-1}{∥p^{i}-A\;\cdot \;\left[\begin{array}{c} m^{i}\\ 1 \end{array}\right]∥}^{2}.$$

This matrix compensates for the overall offset, aligns the axes, and scales the motor steps to millimeters between the coordinate frame of the motor positions and the Cartesian frame of the motion stage. However, the rotating galvos introduce some spherical distortions which cannot be adequately represented by an affine transformation. Therefore, we secondly fit a three-dimensional quadratic function $q:R^{3}\to R^{3}$ to finally obtain Cartesian motor positions ${\hat{m}}^{i}$ as
(10)$${\hat{m}}^{i}=q\left(A\;\cdot \;m^{i}\right).$$

### 2.6. Pose Estimation

We estimate the position of the target in iteration *i* as the centroid of the *N* Cartesian motor positions ${\hat{m}}_{n}^{i}$, n=1,…,N. For this purpose, we resample the positions of the *N* templates to common timestamps. To estimate the orientation in iteration *i*, we apply the Kabsch algorithm [37] to the positions ${\hat{m}}_{n}^{i}$ and the corresponding initial positions ${\hat{m}}_{n}^{0}$ of the templates.

### 2.7. Phantoms and Simulated Motion

As targets for tracking, we consider the phantoms shown in Figure 5. The 40 mm × 40 mm sized plates have a thickness of about 1 mm. They are 3D-printed using photopolymer resin and we receive OCT signals from their backsides as well. The back sides of the plates are rough and randomly structured to simulate sub-surface information. The front side of the first phantom shows the same structure, while the second phantom has a flat surface, instead.

We use a 6-axis robot arm (IRB 120, ABB) to simulate and evaluate different motion patterns. As shown in Figure 6, the end-effector has a sample holder attached to position a phantom about 300 mm in front of the lens of the tracking system. It is positioned such that the axes of its base coordinate frame are approximately aligned to the OCT’s and the tracking system’s axes. This allows one to systematically evaluate two types of rotations. We define in-plane rotations to be in a plane parallel to the OCT’s xy-plane. Therefore, the axis of rotation corresponds to the axial scan direction of the OCT, and hence motion can mostly be compensated by only moving the second pair of galvos while the reference arm length does not have to be adjusted. The other type is out-of-plane rotations, which we define to be in the OCT’s yz-plane.

To evaluate pure translations, we move the robot arm’s end-effector back and forth along a line. For rotations, we rotate in-plane or out-of-plane starting with the original end-effector orientation. Furthermore, we consider random 6D motion. For the sequence of positions, we uniformly draw random points within a cube. For the sequence of rotations, we define a random axis of rotation and uniformly draw a random angle. The positions and the orientations are then realized simultaneously by the robot arm. Each experiment consists of a motion sequence which we record over 45 s. For each repetition, we acquire the templates at a different randomly selected spot on the phantoms before motion starts.

We also use the robot arm to perform the Cartesian calibration. In detail, we move the structured-surface phantom to 80 different positions within a cylinder with a radius of 20 mm and a depth of 40 mm. We only move at 2 mm/s to ensure reliable tracking. At each position, we log the tracking system’s position ten times.

### 2.8. Evaluation Metrics

For pure translations, we evaluate the Euclidean norm of the differences between the relative positions of the robot and those of our tracking system. To evaluate orientations, we use the axis-angle representation, which provides an angle θ which is independent of coordinate systems. Evaluating the six degrees of freedom individually would require one to express translations and rotations in a common frame. While hand-eye calibration could be used to relate the robot and OCT-based tracking system [30,38], it would also be limited by the same system components we use in our setup for tracking, and the residual calibration error would affect the evaluation of the tracking approach. Therefore, we consider the relative rotations in the robot frame and the tracking-system frame independently and compare their magnitudes, which include all three rotational degrees of freedom. Similarly, we established the magnitude of the translation as the Euclidian distance. We mainly consider the root-mean-squared error (RMSE) to describe the results, either with respect to the translations or to the angles.

## 3. Results

### 3.1. Cartesian Calibration

The calibration positions are located at three discrete radii within the cylindric volume, and Figure 7 shows the residual error of the calibration procedure depending on the radius. After fitting an affine transformation to the data points, the residual errors are especially high for the outer positions. After additionally fitting the quadratic function (10), the overall residual RMSE and the maximum residual error are 0.061 mm and 0.136 mm, respectively.

### 3.2. In-Plane Rotations

We consider tracking of rotations and estimation of the current orientation by tracking N=3 spatially separated localitions with the MOSSE filter at the same time. Their spatial arrangement follows Equation (Equation 2) with a radius of r=200 motor steps, to begin with. This corresponds to a radius of approximately 5.4 mm. We first evaluate in-plane rotations using the phantom with a structured surface (Figure 5a). Remember that in-plane rotations mainly require one to move the two galvos but not the stepper motor.

Figure 8 shows that learning rates between 0.01 and 0.03 for the MOSSE filter do not affect tracking accuracy for purely translational motion. For the actual in-plane rotations, we see from Figure 9b that rotations at 4°/s and with maximum angle of 13°/s can be tracked with RMSEs around 0.2° for the lower learning rates. When employing higher learning rates, the RMSEs increase above 0.3°. If we consider rather slow motion at 0.2°/s in Figure 9a, however, the influence of the higher learning rates becomes much clearer. Especially for tracking motion with a maximum angle of 8°, the errors severely increase with increasing learning rate, and Figure 10 shows the deviations between estimated and actual orientations for different learning rates. Using a higher learning rate results in increased underestimation of a slow rotation.

The RMSEs shown in Figure 11 for in-plane rotations of the phantom with a flat surface (Figure 5b)—i.e., without visible superficial information—are similar to the errors for in-plane rotations of the structured-surface phantom. We again observe increased RMSEs for slow rotations when choosing 0.02 for η compared to choosing 0.01. For fast rotation and a maximum angle of 8°, a higher spread of the results compared to the structured-surface phantom can be observed.

### 3.3. Out-of-Plane Rotations

When considering out-of-plane rotations, motion of the target along the OCT’s axial direction will occur. That is, the motorized reference arm has to be used to follow the target by adjusting the reference-arm length accordingly. In previous experiments [33], this mechanical movement appeared to be much more limiting to the dynamic behavior of our system compared to movement of the galvos for lateral compensation.

The results in Figure 12 show larger RMSEs, in the order of 1°, compared to the in-plane rotations. Furthermore, the trackable velocity was more limited. With the lower learning rates, already tracking rotations at 1.5°/s was not successful. In contrast, all reported measurements were successful for η=0.03, for rotation with maximum angles of 3° and of 8°. The reason for the increased RMSE can be seen in the trajectories shown in Figure 13. There is an overestimation of small angles which turns into an almost constant offset for angles above 2°. The general trend, however, is reflected correctly by our tracking approach.

As Figure 14 shows, the errors depend on the radius of the circle on which the templates are positioned. A higher radius provides better estimations. The drawback of larger radii is a smaller range in which motions can be tracked, because we sooner reach the border of the scanning lens. If we combine the larger radius with more than three templates, we can further improve the quality of the estimates. Table 1 shows that the combination of 300 motor steps as the radius *r* and tracking of N=7 templates results in RMSEs in the order of only 0.6°. As a drawback, tracking an 8° rotation already failed at 1°/s in two of three cases. Moving the FOV to the next template’s position requires some time, and therefore, by setting N=7, we increase the time until we evaluate the same template again; i.e., we decrease the effective sampling rate.

As a final proof of concept, we consider estimating the orientation during random 6D motion; i.e., simultaneous translations along all axes and rotations about random axes. We again set r=200 and N=3 as in the first experiments. An exemplary sequence of translations and orientations shows Figure 15. While having an offset as observed before and exhibiting some noise, the estimated angle follows the progress of the actual angle realized by the robot arm. In more detail, Figure 16 shows the RMSEs of the orientations estimated during simultaneous translation at 2, 5 and 8 mm/s within a cube of side length 10 mm. Again, the lower learning rate fails more often for higher velocities at larger maximum angles. However a rotation with maximum angle of 3° at 1.5°/s can be tracked with RMSEs in the order 0.75° while at the same time random translations at 8 mm/s are present.

## 4. Discussion

Conventional optical tracking systems or cameras usually rely on either textured surfaces, or surfaces with varying height, visibility of borders, or artificial markers. In contrast, out proposed tracking approach allows one to track translations and rotations markerlessly, even for the phantom with a smooth surface and no visible borders. We are able to track translations with RMSEs below 0.25 mm (Figure 8). Tracking of in-plane rotations is feasible with errors below 0.3° (Figure 9), even for a large target with a flat surface (Figure 11). By employing OCT, we can exploit micrometer-scale volumetric image data including information about the structure below the surface. Furthermore, imaging is non-invasive and feasible at high temporal resolution.

For our tracking approach using an adaptive MOSSE filter, the choice of the learning rate η can be critical. While increasing η leads to severe underestimation of angles for in-plane rotations (Figure 10), a higher η is necessary for reliable tracking of out-of-plane rotations (Figure 12). This is a general issue, and for an application one should consider which type of motion can be expected. Depending on the velocities expected, either a smaller or a higher learning rate should be chosen. Furthermore, one could consider an approach where the learning rate is adapted online based on the last observed velocities [39].

Tracking of out-of-plane rotations generally performs worse than tracking of in-plane rotations, which only involves the pair of galvos for adjusting the FOV position. The target, however, is still tracked correctly during out-of-plane rotations from a qualitative point of view (Figure 13), even for more complex motion (Figure 15). However, the quantitative evaluation (Figure 12) is distorted by an overestimation of the angles, which occurs throughout all measurements. By arranging the template on a larger circle and by tracking more templates, this effect can be reduced (Table 1). For specific applications, this can, however, lead to deciding for a trade-off between the accuracy and the size of the area in which motion can be tracked.

## 5. Conclusions

In this work, we showed that 6D motion tracking with volumetric OCT is feasible. Our hardware setup allows one to follow motion at multiple locations on a target in order to estimate its position and orientation. Furthermore, we showed that an adaptive MOSSE filter is appropriate to handle the changing appearance of targets in the images caused by rotational motion. Our results indicate that adapting the learning rate with respect to the current speed of the target could be advantageous. Tracking of in-plane rotations was feasible with very promising small errors, even for targets with smooth surfaces. This highlights the value of OCT’s sub-surface information for markerless tracking compared to superficially scanning modalities. Following out-of-plane rotations was feasible as well. However, the task is more tricky, and the motorized reference arm seems to not be an optimal choice. Nevertheless, the trajectories could be qualitatively tracked, and we showed that the quantitative results improved by tracking more locations on a larger radius. Therefore, our proposed concept shows that OCT imaging is promising to realize optical tracking in applications which are difficult for conventional optical tracking systems, especially scenarios not allowing for proper marker placement and providing only access to small and poorly-structured segments of the target.

## Figures and Tables

**Figure 1 sensors-20-02678-f001:**
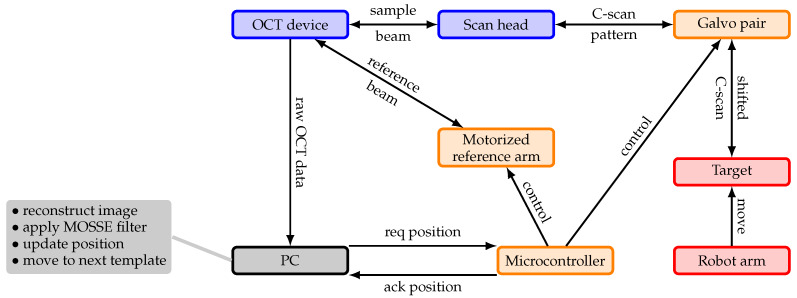
Components of the proposed tracking system. It includes an OCT imaging system (blue), additional hardware to reposition the field-of-view (FOV; orange), a computing device running the tracking loop (gray), and a target moved by a robot arm for evaluation (red).

**Figure 2 sensors-20-02678-f002:**
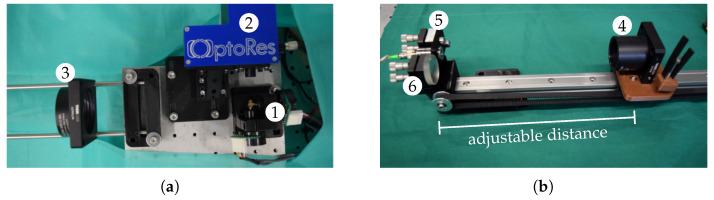
The OCT’s FOV can be shifted laterally by a second pair of galvos (**a**, 1) in front of the actual scan head (**a**, 2) and the range is limited by the achromatic lens with diameter of 50.8 mm (**a**, 3). For axial shifting, a retroreflector (**b**, 4) is mounted to a motor. Due to the setup with the collimator (**b**, 5) and the mirror (**b**, 6), the path length of the reference arm varies as twice the traveling distance of the motor.

**Figure 3 sensors-20-02678-f003:**
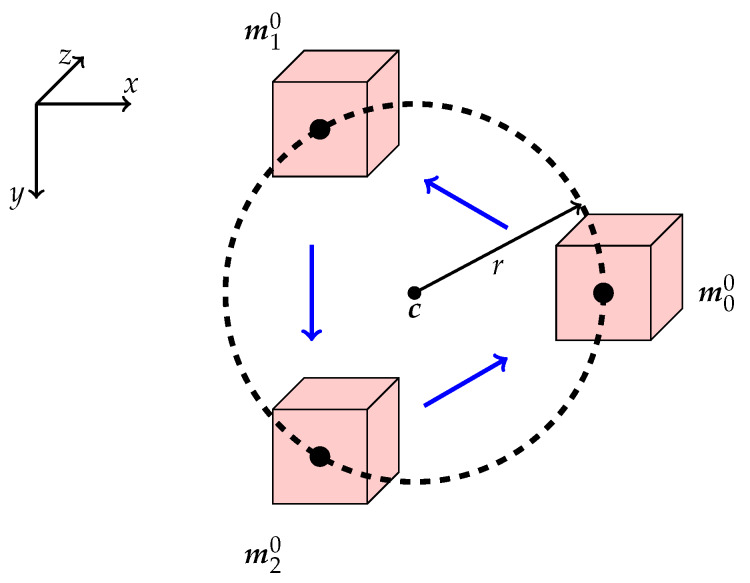
We define, in this example, N=3 templates by moving the OCT’s FOV (red) to three positions $m_{i}^{0}$ on a circle with radius *r*. The initial circle lies within a plane parallel to the xy-plane, but during tracking this might change due to rotational motion.

**Figure 4 sensors-20-02678-f004:**
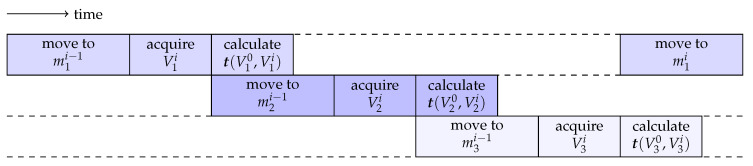
Sketch of pipelined multi-template tracking with three templates. Iteratively, a new volume is acquired and evaluated to determine the motor position for this template in the next iteration. Note that calculating the translations typically is faster than moving to the next position for acquisition.

**Figure 5 sensors-20-02678-f005:**
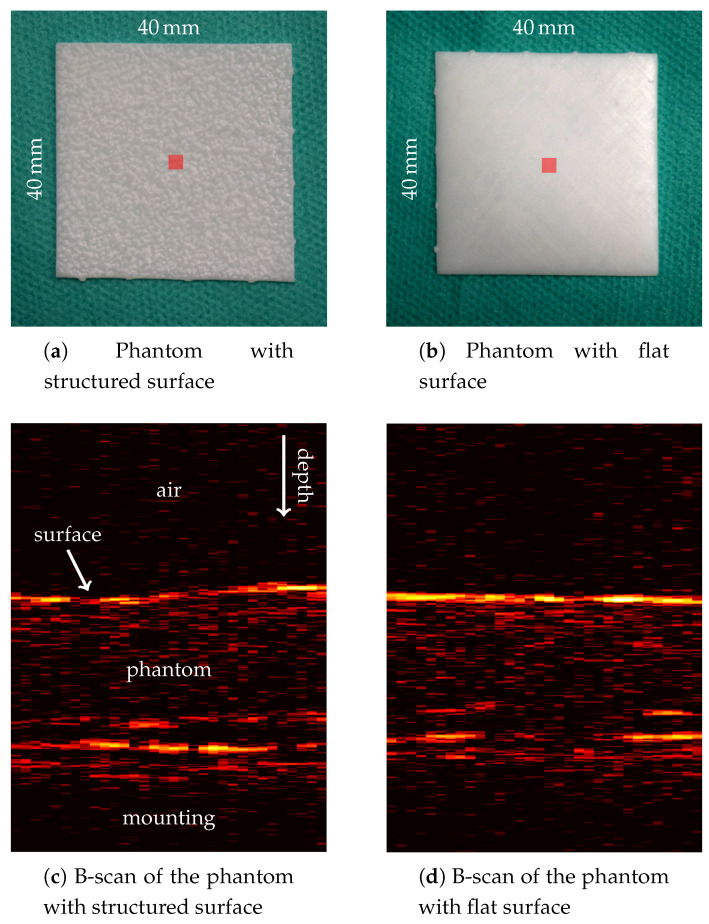
Top row: Pictures of the phantom with a structured surface (**a**) and the phantom with a flat surface (**b**). The backsides of both phantoms are structured too. For comparison, the size of the OCT’s FOV is illustrated as red squares. Bottom row: B-scans of the structured-surface phantom (**c**) and the flat-surface phantom (**d**). Each column corresponds to one A-scan with increasing measurement depth from top to bottom. The B-scans’ pixels are stretched to match their physical dimensions, which are about 2.5 mm laterally and 3.5 mm in depth direction in air.

**Figure 6 sensors-20-02678-f006:**
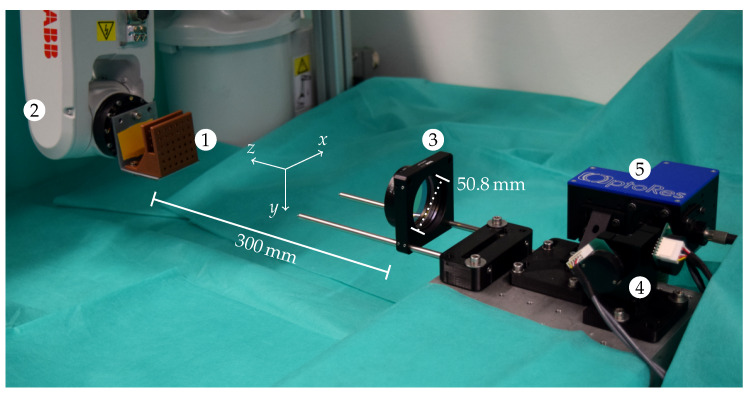
A sample can be mounted to the holder (1) which is attached to the robot arm (2) in front of the scan lens (3), the second pair of galvos (4), and the scan head (5).

**Figure 7 sensors-20-02678-f007:**
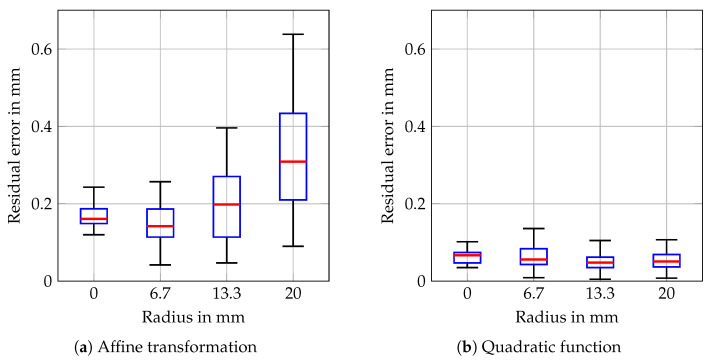
Boxplots of the residual calibration errors after fitting only an affine transformation (**a**) and after additionally fitting a quadratic function (**b**) depending on the radius within the cylindrical volume of the positions. The red bars indicate the medians, the blue boxes range from the 25%-quantiles to the 75%-quantiles, and the whiskers cover all data points.

**Figure 8 sensors-20-02678-f008:**
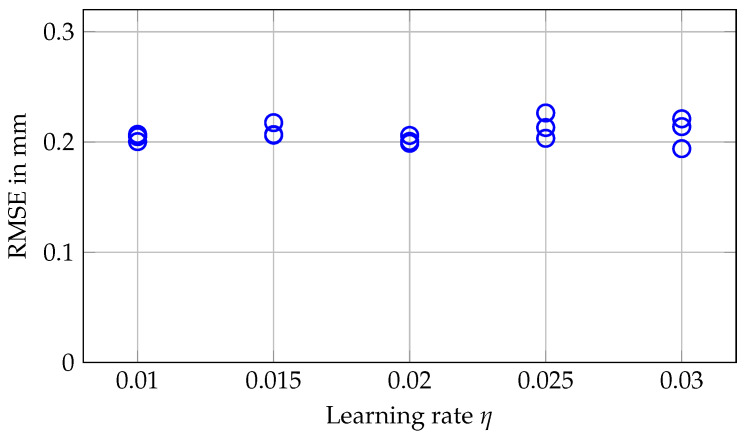
RMSEs for translational motion along all three axes. Three repetitions are shown for each learning rate η. Each time, the robot arm drove back and forth along a diagonal line within a cube with side length 8 mm at a velocity of 12 mm/s.

**Figure 9 sensors-20-02678-f009:**
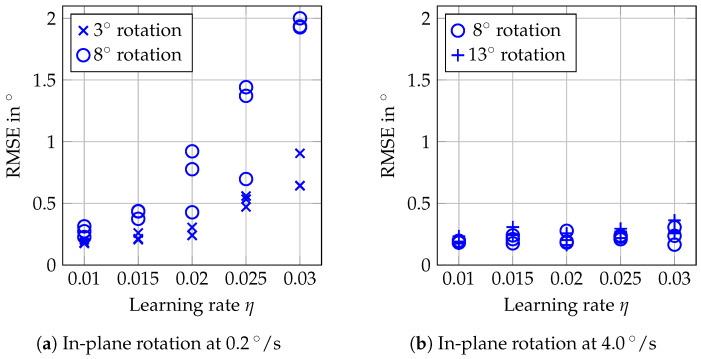
RMSEs for in-plane rotations with the different learning rates η from 0.01 to 0.03 of the MOSSE filter. Different maximum angles were used and the velocities were 0.2°/s (**a**) and 4°/s (**b**).

**Figure 10 sensors-20-02678-f010:**
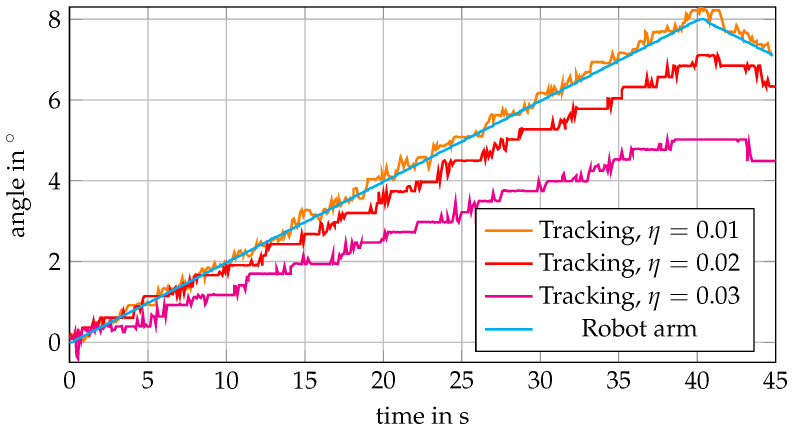
Exemplary curves of estimated angles and robot arm angles for in-plane rotations at 0.2°/s and a maximum angle of 8° (compare Figure 9a). When increasing the learning rate η, the angle is increasingly underestimated.

**Figure 11 sensors-20-02678-f011:**
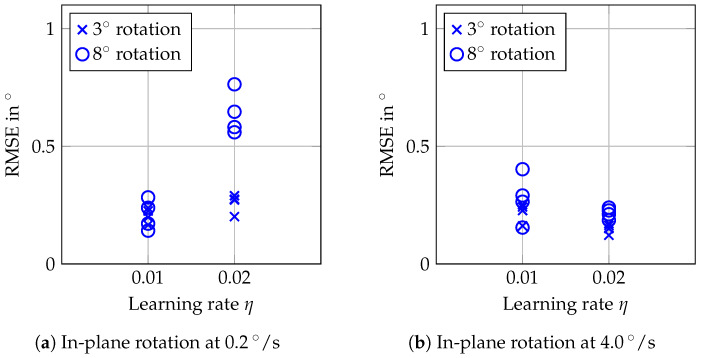
RMSEs for in-plane rotation of the flat-surface phantom.

**Figure 12 sensors-20-02678-f012:**
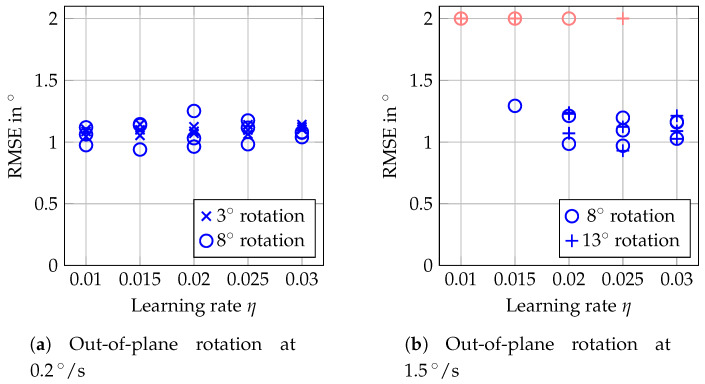
RMSEs of the errors for out-of-plane rotations and different learning rates η from 0.01 to 0.03 of the MOSSE filter. Three repetitions were done for each combination, but RMSEs exceeding 2° are considered as failed tracking and indicated by red marks at 2°.

**Figure 13 sensors-20-02678-f013:**
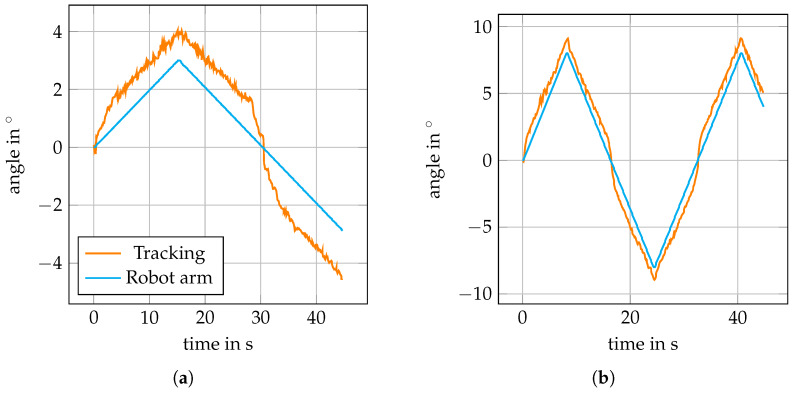
The tracking system (orange) systematically overestimates the robot angles (cyan) for out-of-plane rotations with different maximum angles at 0.2°/s (**a**) and at 1°/s (**b**). Learning rate η=0.02 is used.

**Figure 14 sensors-20-02678-f014:**
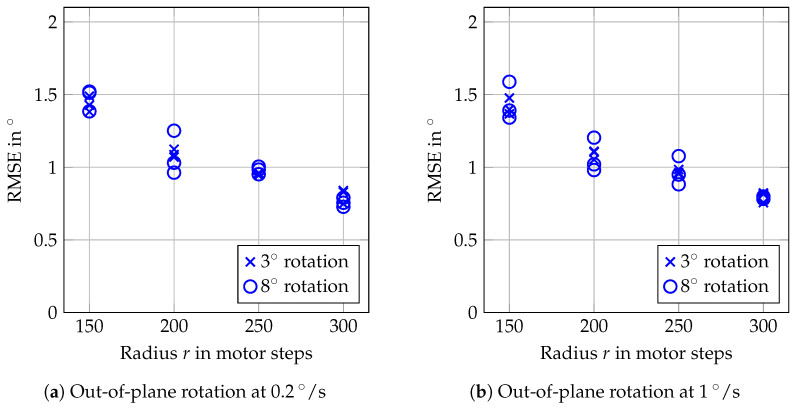
RMSEs for out-of-plane rotations and tracking with different radii *r* in motor steps. The learning rate is set to η=0.02.

**Figure 15 sensors-20-02678-f015:**
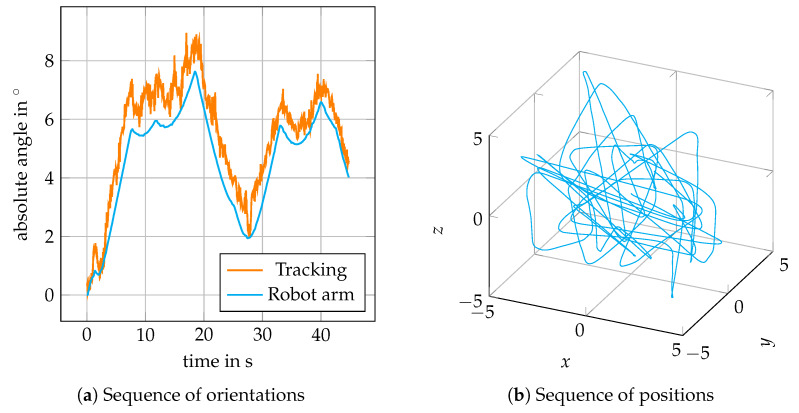
Exemplary trajectory for random 6D motion. The robot arm changes its end-effector orientation randomly at 1°/s up to 8° from its initial orientation (**a**). At the same time, the robot arm translates the phantom randomly within a 10 mm × 10 mm × 10 mm cube at 8 mm/s (**b**).

**Figure 16 sensors-20-02678-f016:**
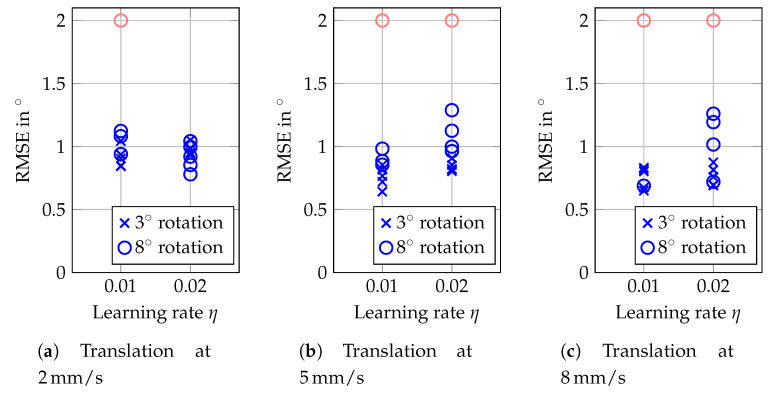
RMSEs for the estimated orientation during random 6D motion. Results are reported for a rotational velocity of 1°/s and translational motion within a cube of side length 10 mm. Results of five repetitions for each parameter combination are shown, but RMSEs exceeding 2° are considered as failed tracking and indicated by red marks at 2°.

**Table 1 sensors-20-02678-t001:** RMSEs for out-of-plane rotations and tracking with parameters N=7, r=300, and η=0.02.

	3° Max Angle		8° Max Angle
	**0.2°/s**	**1.5°/s**	**0.2°/s**
RMSE in °	0.6224 0.6341 0.6423	0.5993 0.6202 0.6347	0.5533 0.6271 0.6281

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
