# Peer review of "Concept for Markerless 6D Tracking Employing Volumetric Optical Coherence Tomography"

_sensors, 2020, doi:10.3390/s20092678_

Round 1

Reviewer 1 Report

Manuscript Review Report for Sensors-762882

Overview Assessment

----------------------------------------------------------------------------
The authors presented a very interesting and useful method for estimating the 6D pose of a rigid target using OCT volumetric data. The method utilizes template-based tracking for multiple locations on the target, and by estimating positions of these templates and by maintaining the correspondence, a pose can be computed in real-time. The manuscript is well written, and the concept is mostly clear.

The reviewer has the following major concerns and suggestions, which are supported by detailed comments in the following sections.

(i) It would be really helpful to have a high-level paragraph at the beginning in 2. Material and Methods to describe the overall technical approach in order to refer to different subsections. And it would be helpful to have a system figure that illustrates each component, such as motor control, single-template tracking, multi-template tracking, and pose estimation, and more importantly, at what rates do they communicate, in one figure.

(ii) The reviewer would like to see the full 6D tracking in the experimental validation, in which currently only orientation was presented. Please see the reviewer’s suggestion on hand-eye calibration.

(iii) The reviewer do not think the experimental validation on orientation is sufficient. Please see the reviewer’s comments regarding in-plane and out-plane rotations.

(iv) The reviewer also suggests in-depth further investigation on orientation error propagation characteristics. Please see detailed comments in below sections.

Comments on Introduction

-----------------------------------------------------------------------------
Text 1. “Second, it realizes fully markerless tracking, i.e., no target-specic information is required.”

Comment 1. From the reviewer’s understanding, it is required to have a successful identification of multiple initial templates. Please clarify if the reviewer understands incorrectly.
Comments on Materials and Methods

Text 2. “The motor and galvos are controlled by a microcontroller which communicates with our measurement PC”

Comment 2. What is the communication/control rate between the motors and the PC?

Text 3. “Given a template C-scan V 0px,y,zq and a later C-scan Vipx,y,zq, an image processing algorithm determines the translation tpV 0,Viq“p∆x,∆y,∆zqT measured in voxels.”

Comment 3. Are there any assumptions or requirements on V 0 such that it is guaranteed to determine a unique corresponding region in Vi?

Text 4. “We consider N templates and their corresponding initial FOV positions m0 n,n “ 1,...N, and arrange them on a circle with radius r around a suitable center point c“px0,y0,z0qT as shown in Figure 2.”

Comment 4. For the tracking problem, is it assumed that the target dose not move while the N samples being taken? If so, then from the reviewer’s understanding, it would require the speeds of FOV positioning much faster then the target motion, would that be fair to say? Please clarify.

Text 5. “We dene in-plane rotations to be in a plane parallel to the OCTs xy-plane. Therefore, the axis of rotation corresponds to the axial scan direction of the OCT and hence motion can mostly be compensated by only moving the second pair of galvos while the reference arm length does not have to be adjusted.”

Comment 5. This seems a confusing definition to the reviewer. The rotational part in the target pose, it is not resulted by sequence of rotations. Therefore, it is just a state, it is an orientation. The reviewer strongly suggest starting with a general case of arbitrary quaternion or rotation matrix. If the authors feel strongly about the in-plane and out-plane, they may describe the projected component of the arbitrary quaternion or rotation matrix. It is not about how you can move the target, but rather how well you can sense the target.

Text 6. “We decide to avoid including the resulting calibration error in our analysis of the system design. Therefore, we only consider the axis-angle representation, even if we also have translations.”

Comment 6. First, “avoid including the resulting calibration error ” is not a good reason to exclude the translations, because the paper is titled 6D and the authors spent a lot of efforts emphasizing on 6D. Second, hand-eye calibration is needed if the authors want to evaluate relative pose of target in OCT frame. But if the authors define an “anchored” using the initial pose, and use the relative pose of estimate w.r.t. the “anchored” ones in both robot and OCT frames, respectively, then the authors could evaluate the full 6D tracking consistency by applying some similarity transformations.

Comments on Results

----------------------------------------------------------------------------
Text 7. “3.2 Out-of-Plane Rotations”

Comment 7. It is not clear how the out-of-plane rotations of the rigid target motion is generated. Does the rotation axis keep changing when moving the target? In order to test the orientation tracking performance, the reviewer thinks a good generalized validation would be to keep changing the rotation axis while changing the rotation angle. Again, this goes back to the definition of an arbitrary orientation should be independent how it is rotated to arrive.

Comment 8. The reviewer suggests investigating into the orientation error directions in addition to only the magnitudes (angles in axis-angle). The error angle in axis-angle representation gives the magnitude information, but, to understand deeper regarding the error propagation characteristics, the reviewer suggests that the authors adopt Bingham distribution representation to investigate orientation errors.

Reviewer 2 Report

The authors have proposed a markerless optical tracking scheme in optical coherence tomography. The proposed scheme was demonstrated using a phantom. The submitted paper can be accepted provided following comments are addressed

  • In the introduction, part authors have mentioned a few previous works on wide FOV imaging in OCT. Parallel imaging is another way to achieve wide-field imaging in OCT, they should mention that technique as well, for example, the reference below

"Yongyang Huang, Mudabbir Badar, Arthur Nitkowski, Aaron Weinroth, Nelson Tansu, and Chao Zhou, "Wide-field high-speed space-division multiplexing optical coherence tomography using an integrated photonic device," Biomed. Opt. Express 8, 3856-3867 (2017)"

  • The OCT system description is not clear they only mention OMES, Optores system was used. The mentioned OCT system consists of three modules laser, a scanning head, and an imaging module. The authors used all three of them or only laser? There should be more details on the OCT setup and also about acquisition such as how nonlinear tuning was corrected
  • What is scale bar for fig. 4 (c) and (d)
  • can they emphasize on which OCT application will advantage from their proposed tracking scheme and how
